



# Sensitivity of calving glaciers to ice-ocean interactions under climate change: New insights from a 3D full-Stokes model

[1,2]Joe Todd, [1]Poul Christoffersen, [3]Thomas Zwinger, [3]Peter Råback, [2]Douglas I. Benn

[1]Scott Polar Research Institute, University of Cambridge, Cambridge, UK,
[2]Department of Geography and Sustainable Development, University of St Andrews, St. Andrews, UK,
[3]CSC-IT Center for Science, Espoo, Finland

*Correspondence to*: Joe Todd (jat39@st-andrews.ac.uk)

**Abstract.** Iceberg calving accounts for between 30-60% of net mass loss from the Greenland Ice Sheet, which has intensified and is now the single largest contributor to global sea level rise in the cryosphere. Changes to calving rates and
the dynamics of calving glaciers represent one of the largest uncertainties in projections of future sea level rise. A growing body of observational evidence suggests that calving glaciers respond rapidly to regional environmental change, but predictive capacity is limited by the lack of suitable models capable of simulating the calving mechanism realistically. Here, we use a 3D full-Stokes calving model to investigate the environmental sensitivity of Store Glacier, a large outlet glacier in West Greenland. We focus on two environmental processes: undercutting by submarine melting and buttressing by ice
mélange, and our results indicate that Store Glacier is likely to be able to withstand moderate warming perturbations in which the former is increased by 50% and the latter reduced equivalently. However, severe perturbation with a doubling of submarine melt rates or a complete loss of ice mélange destabilizes the calving front in our model runs. Furthermore, our analysis reveals that stress and fracture patterns at Store's terminus are complex and varied, primarily due to the influence of basal topography. Calving style and environmental sensitivity varies greatly, with propagation of surface crevasses
significantly influencing iceberg production in the northern side, whereas basal crevasses dominate in the south. Any future retreat is likely to be initiated in the southern side by a combination of increased melt rate in summer and reduced mélange strength in winter. The lateral variability, as well as the importance of rotational and bending forces at the terminus, underlines the importance of using the 3D full-Stokes stress solution when modelling Greenland's calving glaciers.

## 1 Introduction

Frontal ablation of tidewater glaciers (calving and submarine melt) is the most important ablation mechanism in the cryosphere, accounting for around half of the net annual ice loss in Greenland over the last decade (Van den Broeke et al. 2009) and more (up to two-thirds) when warm subtropical waters periodically flow into coastal seas and fjords (Rignot and Kanagaratnam, 2006; Holland et al., 2008; Straneo et al. 2010; Christoffersen et al. 2011). The remaining loss is caused by surface melting (Enderlin et al. 2014; van den Broeke et al., 2016). In Antarctica, ice sheet mass loss is partitioned evenly
between calving flux and ablation tied to basal melting of large ice shelves (Depoorter et al., 2013, Rignot et al. 2013).

The importance of calving as a contributor to global sea level rise is demonstrated through the sustained acceleration, and subsequent dynamic thinning of glaciers exposed to environmental forcings at the ice-ocean interface where calving occurs (Howat et al., 2005, Holland et al., 2008,  Rignot and Kanagaratnam, 2006). Recent observational studies (James et al., 2014,
Murray et al., 2015, Chudley et al., 2018, Medrzycka et al., 2016, Luckman et al. 2015) have captured the calving process in unprecedented detail, yet the physical links between calving and climate remain poorly understood.

Several environmental and internal factors are hypothesised to affect calving, which occurs when fractures propagate through glacier termini in response to growing stresses in the ice, leading to the detachment of icebergs (Fig. 2).





Undercutting of the terminus by submarine melting (Fig. 2a) has been suggested as an important environmental stimulus for calving (O'Leary and Christoffersen, 2013, Luckman et al., 2015); this ice-ocean interaction is amplified by fresh glacial meltwater (Fig. 2e) which is discharged subglacially into fjords, forming forced-convective plumes which entrain ambient water and can melt the terminus at rates of up to several metres per day (Jenkins, 2011; Xu et al. 2012, Slater et al., 2016).

Submarine melt rates of such high magnitude are a consequence of warm ambient fjord waters and large quantities of supraglacial meltwater routed along the bed (Jenkins, 2011; Slater et al., 2016), linking this process to both oceanic and atmospheric conditions (Christoffersen et al., 2012). Another environmental factor is ice mélange (Fig. 2b), the rigid mixture of calved icebergs and sea ice often found in front of Greenland outlet glaciers in winter and spring, which may suppress calving by providing a buttressing force on the glacier terminus (Sohn et al., 1998; Amundson et al., 2010, Walter et al.

2012). Modelling studies suggest this effect may be sufficient to promote seasonal terminus advance (Vieli and Nick, 2011; Todd & Christoffersen, 2014, Todd et al., 2018) and observational studies show that seasonal and interannual glacier retreat is often concurrent with pro-glacial mélange disintegration (Howat et al., 2010, Moon et al., 2015, Bevan et al., 2019).

The internal dynamics of outlet glaciers strongly modulate the effect of external forcing on calving (Moon & Joughin, 2008,

Schild & Hamilton, 2013). The fact that nearby glaciers may respond differently to the same regional-scale climatic shifts is evidence of this. Bedrock topography (Fig 2c) strongly influences glacier dynamics, providing both basal and lateral pinning points, both of which tend to define stable terminus positions (Benn et al., 2007a). Related to this, spatial and temporal variability in basal friction (Fig 2d), which is a function of subglacial hydrology as well as glacier geometry, has been suggested as a potentially first-order control on calving (Benn et al. 2007a).

A range of approaches has been developed to represent calving processes in numerical models (e.g. Nick et al., 2010, Cook et al., 2014, Todd et al., 2014, Krug et al., 2014; Morlighem et al., 2016). These typically adopt simplified representations of stress and calving processes, and are generally tuned against observations rather than independently validated. Recently, calving has been incorporated into a 3D full-Stokes model (Todd et al., 2018), allowing calving processes to be simulated in

much greater detail than previously possible. In this study, we use this model (Elmer/Ice) to investigate the environmental sensitivity of Store Glacier (Fig. 1), a large outlet glacier flowing into Uummannaq Bay in West Greenland. Store Glacier (henceforth referred to as 'Store') is 5 km wide at the terminus, where ice velocity reaches 20 m d$^{-1}$. The glacier has remained in a stable position for at least the past 40 years (Howat et al., 2010), through periods when several nearby glaciers have undergone concurrent retreat. While interannually stable, Store's terminus displays a distinct seasonal advance and

retreat cycle of up to 1km; Todd et al. (2018) were able to reproduce the observed calving seasonality of Store Glacier by applying only two types of environmental forcing: undercutting of the terminus by submarine melting and buttressing from seasonally rigid ice mélange. That study found buttressing by ice mélange to be the primary driver of observed seasonal advance and retreat, while submarine melting prevents the terminus from advancing beyond its current stable position.

Here, we extend the analysis of Todd et al. (2018) by investigating the sensitivity of iceberg calving at Store, with the aim of understanding how the glacier will respond to future changes in environmental forcings associated with a warming climate. With calving simulated over multiple years and with individual calving events resolved in 3D, we present an analysis which is unprecedented in detail. We show that a doubling of submarine melting at Store Glacier's terminus would likely lead to accelerated calving and retreat, predominantly in the floating southern half of the glacier's terminus. We find that the

*distribution* of submarine melting is critical to the glacier's stability, and that the loss of winter/spring ice mélange buttressing could have knock-on effects on summer stability. We also find that subglacial topography is critical in determining calving dynamics and glacier stability.



## 2 Methods

To determine climatic influences on Store Glacier's calving, ice flow, and stability, we perform sensitivity experiments in which calving in our model responds to changed environmental conditions. The starting point is a present-day control simulation which includes three types of climate forcing on the terminus: 1) distributed submarine ice-wall melting at the submerged portion of the terminus, 2) concentrated submarine melting associated with convective plumes forming at two known locations where glacial meltwater is subglacially discharged, and 3) ice mélange buttressing.

### 2.1 Environmental Forcing in Control Simulation

Submarine ice-wall melting occurs when cold, fresh, buoyant subglacial meltwater is discharged into warm and dense fjord water, creating a forced convective plume (Fig. 2b); melt rates are linked to rate of subglacial discharge, and the temperature and salinity of the fjord water (Xu et al., 2013, Jenkins et al., 2011, Slater et al., 2015). Patterns of submarine ice-wall melting are complex (Chauché et al., 2014, Chauché, 2016, Rignot et al., 2015), but can be decomposed into planar plume melting, driven by distributed subglacial discharge across the entire front, and localised concentrated plumes from highly channelised discharge. As in Todd et al. (2018) we investigate the effects of the undercutting caused by these end-members.

Distributed melting of the submerged portion of the terminus is applied year round, using vertical melt rate profiles derived from an analytical plume model (Slater et al., 2016) and temperature-salinity records acquired throughout the year from the fjord in front of Store (Chauché et al., 2014). The vertical melt rate profiles are applied uniformly across the ice front, but with higher rates of melting in summer than winter. For the control experiment the front-averaged melt rates are 1.3 m d$^{-1}$ for winter (September to May) and 3.1 m d$^{-1}$ for summer (June, July and August).

Concentrated plume melting occurs in summer (JJA) when runoff from the surface is routed along the bed and discharged subglacially into the fjord through large channels. In winter (all other months) we assume that the subglacial hydrology consists of a wholly distributed system. Concentrated plumes are applied as conical melt features that expand as they rise, reaching a radius of 162m at the surface. Concentrated melting is included at two known locations along the calving front (Fig 3b) where plumes are frequently observed as sediment-laden water reaching the surface of the fjord in summer. As with distributed melting, we use the analytical plume model to determine vertical melt profiles for the two conical plumes, although in this case we prescribe a *maximum* in-plume melt rate (12 m d$^{-1}$ in the control experiment). While the maximum rate of concentrated melting is high relative to that of distributed melting, concentrated melting is spatially confined to a relatively small portion of the submerged ice front (3%), whereas distributed melting covers the entire ice wall and therefore melts more ice.

The third environmental forcing is proglacial ice mélange, which forms a seasonally rigid mixture of icebergs, bergy bits and sea ice in front of most Greenlandic outlet glaciers, including Store Glacier (Howat et al., 2010). This rigid ice mélange provides a buttressing force on the glacier terminus which is hypothesised to suppress calving (Howat et al., 2010, Walter et al., 2012, Todd et al., 2018) by preventing fracture and iceberg detachment. Ice mélange buttressing is implemented as an external pressure on the calving front, similar to the implementation of water pressure from the sea. In the control experiment we use a value of 120 kPa from Todd et al. (2018) and we apply this force uniformly over a thickness of 140m starting 1st February and ending 29th May. Thickness is based directly on observations (Todd et al., 2018), while back pressure is based on force balance analysis similar to the approach used by Walter et al. (2012).



## 2.2 Sensitivity Analysis

In the sensitivity experiments, we examine changes to all three types of environmental forcing, as well as their combined effect (Table 1). We run 5-year simulations in which we modify the magnitude of these environmental processes, consistent with an ongoing trend of warming ocean and atmosphere, in order to test the glacier's short-term response to these

perturbations. Distributed and concentrated melt rates are scaled separately by a factor of 1.5 (Runs MD1, MC1) and 2.0 (Runs MD2, MC2) in order to test the glacier's response to 50% and 100% increases. For the distributed plume experiments (MD1, MD2), we re-calculated the analytical melt profiles to match the new, scaled average melt rate. For conical plume melting (MC1 and MC2), we found it difficult to achieve the desired scaling with the conical plume model, due to the nonlinear relationship between subglacial discharge and melt rate. Thus, we instead directly scale the present-day conical

plume melt profile from the control experiment.

We also scale the ice mélange thickness by 0.5 (Run MM1) and 0 (Run MM2), scenarios which reflect halving thickness and a complete loss of the mélange, respectively. Finally, we combine the scaling of all three forcings in intermediate (Runs MA1) and severe (Runs MA2) perturbation scenarios. We also run a control simulation (Run CONTROL) in which the

environmental forcings are identical to those in Todd et al. (2018). All of the sensitivity experiments outlined above were run for 5 years, or until the simulation broke down irrecoverably.

## 2.3 3D Calving Model

We use Elmer/Ice's 3D calving implementation, presented in Todd et al. (2018), to investigate Store Glacier's sensitivity to changes in environmental forcing. The model implements the crevasse depth calving criterion (Benn et al. 2007b, Nick et al.

2010), modified to take advantage of the full-Stokes stress solution and third dimension (Todd et al. 2018). Calving is predicted to occur either when surface and basal crevasses meet, separating an iceberg from the terminus, or when surface crevasses reach the waterline, which is assumed to lead to hydrofracturing and full crevasse penetration. Crevasse penetration is computed using a zero stress condition (Nye, 1957), such that open crevasses exist wherever a net tensile stress exists in some direction. We define the effective principal stress (EPS) for both surface and basal crevasses:

$$EPS_{surf} = \sigma_1 \tag{1}$$

$$EPS_{basal} = \sigma_1 + P_w \tag{2}$$

where $\sigma_1$ is the largest principal stress and $P_w$ is water pressure acting to open basal crevasses. Water pressure in basal crevasses near the terminus is taken equal to that in the proglacial fjord, assuming an efficient drainage system links the subglacial hydrology to the fjord. The calving algorithm, as well as the model's remeshing and time-evolution, are described

in detail and discussed in Todd et al. (2018).

## 2.4 Domain and boundary conditions

Model domain and boundary conditions are identical to Todd et al., 2018, aside from changes to the climate forcings described below. The model domain extends 112 km inland, and is laterally constrained by Store's ice-flow catchment (Fig. 1a). We solve the full-Stokes equations for ice flow using Glen's flow law (Cuffey and Patterson, 2010). On the inflow

boundary, velocity is set equal to mean observed velocity from TerraSAR-X imagery (April 2014-April 2015). A no penetration condition is imposed on the lateral boundaries, with slip coefficient of 1.0 x $10^{-3}$ for ice/ice interfaces and 1.0 x $10^{-2}$ for ice/rock interfaces. A similar condition applies on the base of the ice, but inverse methods (Gillet-Chaulet et al., 2012) are used to determine basal slip coefficient ($\beta$), which varies through the season, in order to reproduce observed seasonal patterns of velocity. Where the model predicts that ice is floating, basal traction disappears ($\beta = 0$) and the bottom





surface is permitted to lift off the bed. On the calving front, external pressure from the fjord water is applied to ice below sea level. Ice above the waterline is stress free, as is the upper ice surface. Evolution of the upper ice surface is computed throughout the simulation, and surface mass balance is applied as an annual average, determined from RACMO data (Noël et al., 2015).

**2.5 Model Spin-up**

The simulations presented here run from the end of the 5-year 'present-day' simulation presented in Todd et al. (2018), which in turn runs from the end of a 300-year spinup in which the calving front is held fixed and the ice temperature and upper and lower ice surfaces are allowed to evolve. Ice temperature is then held constant during the calving simulations for reasons of computational efficiency. This is justified because significant changes in the temperature field are unlikely to occur over such short timescales. For the computation of the temperature field, ice temperature on the upper surface and inflow boundary are set using the MODIS IST product, averaged from 2000-2014. Geothermal heating is applied on the base at 75 mW m$^{-2}$ (Greve, 2005), and basal frictional heating and internal strain heating are also accounted for using Elmer/Ice's temperature subroutines. Ice flow and temperature are linked by the temperature dependent rate factor of Glen's flow law.

**3 Results**

**3.1 Control simulation**

The control simulation in this study represents present-day environmental forcing, as in Todd et al. (2018). In this present-day baseline scenario, Store's terminus displays a seasonal range which varies greatly from north to south (Fig 3), with greater seasonal range in the south. Seasonal advance begins in February, and up to the end of May the terminus advances by 500m on average, followed by a rapid retreat as soon as mélange is removed (Fig 4). The terminus then maintains a fairly stable position through the rest of the year, varying stochastically by around 100m on average. With the exception of the 200m retreat at 1.6 years (Fig 4), terminus position shows no sensitivity to increased melting in summer.

**3.2 Response to intensification of distributed submarine melting**

We find that a 50% increase in distributed melting at the calving terminus (Run MD1) has only a small effect on the seasonal evolution of the modelled terminus geometry (Fig. 3a). The span of the maximum seasonal advance in late spring and retreat in summer is around 750 m, as opposed to 600 m for the control run (Fig. 4a), and the seasonality of advance and retreat remains periodic and similar to the control experiment throughout the simulation.

Doubling the distributed melt rate (Run MD2) has, however, a significant and destabilising effect on the terminus geometry (Fig. 3a). For the first model year of this simulation, the mean terminus position is very similar to that of Run MD1 (Fig. 4a). However, at the start of the summer melt season in model year 2, the terminus rapidly retreats by 900 m on average. This retreat is far greater than that in the control simulation and the perturbation triggers a transient increase in the terminus velocity from around 5200 m a$^{-1}$ to 5700 m a$^{-1}$, lasting 4 days (Fig. S3a). Just over a month later, the terminus calves back another 470 m on average. Fig. 3a shows that the additional calving retreat occurs in the southern half of the terminus, where the glacier is afloat, and the terminus retreats by up to 3 km in just 40 days (Fig 5b). The rapid retreat causes the simulation to break down irrecoverably as the free surface evolution in our model cannot accommodate geometry changes of this magnitude.

A calculation of the mean annual terminus mass budget components reveals that the calving rates in Runs MD1 and MD2 are less than that in the control simulation (Table S3). In Run MD1, the 50% increase in distributed melting decreases the



average annual mass loss from calving from 7.2 Gt to 6.7 Gt. Doubling the distributed melt rate (Run MD2) causes calving rate to increase slightly compared with MD1 to 7.07 Gt yr$^{-1}$, due to the collapse of the southern side of the terminus. Importantly, the reduction in calving mass loss is more than offset by the increase in loss through distributed melting.

### 3.3 Response to intensification of concentrated submarine melting

Increasing the melt rate from concentrated plumes by a factor of 1.5 (Run MC1) has no systematic effect on the shape of the terminus (Fig. 3b) nor its velocity (Fig. S3b) over the 5 year simulation. There are some minor differences in the minimum extent of the terminus, but the pattern varies between years. However, doubling the concentrated melt rate (Run MC2) has a significant effect on the retreat of the terminus in summer (Fig. 4b). In every year, the terminus retreats further than the control simulation; in year 1 the minimum mean terminus position is almost 600 m further upstream than the control. Figure

3b shows that, as with distributed melting, this extra retreat occurs exclusively in the floating region in the southern half of the terminus. In fact, the region of the terminus which retreats further than the control is constrained on either side by the two concentrated plumes. No significant changes in terminus velocity accompany the additional retreat.

### 3.4 Response to reduced ice mélange buttressing

While the seasonal formation and collapse of rigid proglacial ice melange is the primary control on the seasonal advance and
retreat of the terminus (Todd et al., 2018), we find that reducing the ice mélange thickness to half (Run MM1) has only a limited effect on seasonal position (Fig 3c). In the winters of years 3-5 the terminus advanced slightly less than the control, and in the summer of year 1 the terminus retreated around 150m further upstream (4c). However, the broad pattern of both terminus range (Fig 3c) and seasonal evolution (Fig 4c) was much the same as the control. The deceleration of the terminus during the mélange season (300 m a$^{-1}$) was less than the control simulation (650 m a$^{-1}$), due to the reduced mélange thickness
(Fig S3c).

Removing the mélange entirely has a significant effect on the seasonal range in terminus position. The characteristic spring advance of the floating tongue is absent (Fig. 4c), and in year 4 the terminus retreats 350 m further than the control simulation, implying that the mélange has an important stabilising effect on timescales longer than a year. Again, this retreat
occurs in the floating southern region of the terminus.

### 3.5 Response to combined intensified forcing

When all three forcings are applied together we find limited sensitivity in Run MA1 where both types of melting were increased by 50% and the melange thickness reduced to half. The mean terminus position remains quite consistently around 150 m upstream of the control simulation (Fig 4d). The advance due to ice mélange occurs more slowly, and increased
melting keeps the terminus in a slightly retreated position compared to the control. As with previous magnitude perturbation experiments, the retreat is mostly confined to the southern side of the terminus (Fig. 3d).

In Run MA2, when submarine melting is doubled and mélange is entirely absent, the behaviour of the terminus is drastically different compared to run MA1 as well as the control. The terminus immediately begins a gradual retreat, which accelerates
during the first summer melt season, retreating 450m in total (Fig 4d). There is a gradual readvance of 400m in the following winter, but the glacier rapidly retreats by 800m in the following summer. The terminus remains in this retreated position until the end of August, after which the simulation breaks down due to further calving which causes problems for the model's remeshing algorithm. Figure 3d shows that this retreat is confined to the southern, floating portion of the terminus, the same region as was observed to retreat in all the other magnitude forcing simulations. In this case, the entire southern side of the
terminus has retreated more than 1 km beyond the minimum position observed in the control simulation. The velocity of the





terminus in Run MA2 is fairly consistent with the control, with two exceptions. Firstly, the lack of mélange means there is no terminus slow-down in spring. Secondly, for half a year following the start of the dramatic retreat, the terminus velocity was at least 300 m a$^{-1}$ faster than the control, after which the simulation breaks down.

As an alternative, we also tested model sensitivity to changes in the duration of the applied forcings, but this had an overall limited effect on calving and the stability of the glacier's terminus position (See supplementary text S1).

## 4 Discussion

Our results indicate that Store Glacier should remain relatively unchanged under intermediate scenarios of climate change, but that it may undergo retreat if climate change becomes severe. We base these assertions on novel model simulations in
which a 50% increase in the rates of submarine melting had overall limited effect on glacier dynamics, whereas a doubling led to retreat of more than a kilometre compared with the control, with (Figs. 3a, 4a) or without (Figs. 3d, 4d) mélange buttressing. Retreat in these simulations led to breakdown of the model due to rapid changes in the shape of the domain. At present, we are unable to speculate as to whether this retreat would continue or if the terminus would stabilise. Velocity at the terminus in Run MA2 shows a sustained acceleration relative to the control (Fig. S3d) which may indicate instability, but
further model developments are required to simulate the longer term evolution of the calving front. In addition to the effect of submarine melting, we found that the absence of winter mélange not only prevents the annual advance of the terminus, but also has a knock-on effect on the calving rates in the following summer.

We also found topographic setting is a critical control on calving and terminus stability. In previous work, the vertically-
integrated Ice Sheet System Model (ISSM) has been used with a calving rate parameterisation to investigate the effect of submarine melting on calving at Store Glacier (Morlighem et al. 2016). This analysis found no significant effect on terminus position from a doubling of submarine melting to 6 m/day, in contrast to our own results. This is likely because vertically-integrated models cannot directly simulate the stress effects of melt undercutting. Interestingly, in ISSM both the 'control' and 'double melt' simulations retreat to a steady state which looks qualitatively similar to our MD2 minimum position (Fig.
3a) just before our simulation breaks down. This similarity in retreat pattern underscores the importance of basal topography in controlling calving: two very different ice flow models equipped with different calving laws show similar spatial patterns of retreat, though in response to different forcing. However, the inability of the ISSM model to reproduce the present-day terminus position highlights the need for fully 3D calving models equipped with physically-based calving laws.

The sensitivity analysis performed here emphasises the complex relationship between submarine melting and calving. The potential destabilising effect of submarine melting depends on the distribution of the melt forcing, and appears significantly greater in the southern side of the terminus than in the north, due to the effect of topography, which we discuss below. In the short term, Store Glacier appears to be able to withstand a 50% increase in submarine melt rate compared to our control, which represents present-day conditions (Figs 3,4). Given that our simulations last 5 years, we cannot, however, rule out the
possibility that a 50% increase in submarine melt rates could lead to changes in terminus geometry over longer periods. The model seems relatively insensitive to an increase in the *duration* of submarine melting (Figs S1,S2), suggesting that longer summers would not necessarily destabilise Store. This relative insensitivity contrasts with previous studies which suggest that submarine melting has a significant effect on glacier calving rate and terminus stability (Rignot et al. 2010, Luckman et al. 2015). O'Leary and Christoffersen (2013) used a simple idealised 2D model to demonstrate that undercutting by
submarine melting might increase tensile stress at the ice surface, potentially opening crevasses farther from the front and thus promoting calving events. The reason that Store Glacier is relatively insensitive to intensification of submarine melting



is two-fold. One factor is the glaciers' fast flow, which advects ice across the grounding line much faster than the rate of undercutting, and the other is the glacier's topographic setting. Below, we discuss both aspects in more detail.

Insofar as stability is concerned, we find that a 100% increase in distributed submarine melting is required to induce a purely undercut-driven retreat of Store Glacier (Fig 5b). However, the modelled terminus also underwent significant seasonal retreat in response to increased *concentrated* submarine melting. Concentrated submarine melting cuts notches into the terminus, isolating the surrounding ice from the sidewalls, reducing lateral drag and promoting calving. Plume modelling studies (Xu et al. 2013, Kimura et al. 2014, Slater et al. 2016) indicate that distributed plume systems melt ice more efficiently than localised concentrated plumes, but our simulations show that concentrated plumes have the potential to destabilize large portions of the calving terminus despite their relatively small contribution to the total melt of the terminus.

Rates of submarine melting by forced convective plumes increase linearly with fjord water temperature and sublinearly with subglacial discharge (Xu et al., 2013), such that doubling the *distributed* submarine melt rate would require a doubling of thermal forcing (fjord water temperature), a quadrupling of subglacial discharge, or some combination of the two. However, a reorganisation of the subglacial hydrology to a more channelised efficient system could promote more concentrated melting at the expense of distributed melting; this could enhance calving without necessarily requiring a large increase in total subglacial discharge or fjord water temperature. Surface velocity records from Store Glacier (Ahlstrøm et al., 2013) show a distinctive late-summer slow-down, similar to that which previous authors have attributed to a switch in the subglacial hydrological system from distributed to channelised subglacial drainage and back (Bartholomew et al., 2010; Sundal et al., 2011). Thus, if a warming climate were to exert an influence on the timing and characteristics of this switch, the resulting change in the subglacial hydrological system could enhance concentrated plume melting and, thus, calving.

Todd et al. (2018) found that ice mélange was the main driver of seasonal terminus advance at Store under present day climatic conditions, and our sensitivity analysis confirms this will continue as long as mélange is present (Figs. 3c,4c). Mélange acts to stabilize the calving front by transferring buttressing force from fjord sidewalls through granular jamming (Peters et al., 2015, Robel, 2017). This buttressing force is applied to the terminus around the waterline, inhibiting surface crevasse formation and preventing bergs from overturning. We nonetheless found only a limited response in glacier behaviour when ice mélange thickness was halved (Fig. 3c), suggesting that proglacial ice mélange will continue to promote terminus advance by closing crevasses, even if thickness or strength is reduced. When mélange was completely removed (Figs. 3c, 4c), the floating tongue fails to advance, corroborating earlier work (Todd and Christoffersen, 2014). However, the model also undergoes significantly greater retreat in summer compared to the control, with mean terminus position retreating 400 m further upstream than the control, despite no additional melt forcing being applied. This implies that mélange buttressing is not simply an instantaneous stabilising force, but also promotes longer term interannual stability. Observations from other glaciers in the Uummannaq region (Howat et al. 2010) support this hypothesis: in the spring of 2003, three glaciers failed to undergo their usual spring advance, and subsequently began concurrent multi-year retreat. We propose that terminus advance and reduced calving in winter, driven by mélange buttressing, contributes to interannual stability through positive 'dynamic' mass balance; as the terminus advances, increased basal and lateral drag modifies the force balance, leading to deceleration and dynamic thickening. This counteracts the dynamic thinning and mass loss which occurs in summer.

Our sensitivity analysis suggests that Store's calving response to environmental forcing is both complex and heterogeneous. The results indicate that overall, Store is relatively insensitive to the changes associated with intermediate scenarios of climate warming. Under more severe warming, the model shows much greater environmental sensitivity in the southern side




of the terminus than in the north. Modelled retreat was especially pronounced in the southern half of the terminus, where enhanced calving creates a new embayment, while the north half remains relatively unchanged (Fig 5b). We argue that Store's relative stability and the markedly different response of the two sides of the glacier are evidence of the strong control exerted by topography on the relationship between calving and climate forcing.

Calving glaciers tend to terminate in regions of topographic stability (Warren and Glasser, 1992, Benn et al., 2007a, Post et al., 2011, Catania et al., 2018) and this is particularly true for Store Glacier. Store's terminus is located at a narrow point between two mountain ridges, beyond which the fjord widens significantly (Fig. 1b). Although bed elevations at the terminus are around -400 m.a.s.l., a subglacial trough deepens inland to depths of -1,000 m.a.s.l. Thus, Store's terminus is at a local

maximum in terms of both basal and lateral resistive stress. Ice flowing towards the terminus from upstream will experience rapidly increasing resistive stress, which suppresses calving. As the ice passes the lateral and basal pinning points, this resistive stress is rapidly lost, promoting calving. Thus, Store Glacier is characterised by the rapid delivery (16-20 m d$^{-1}$) of ice to a terminus position where calving is inevitable; the system is akin to a conveyor belt dropping icebergs into Uummannaq Fjord. This setting contrasts strongly with several Svalbard glaciers where calving rates are observed to vary

linearly with ocean temperature (Luckman et al., 2015). There is no information about the submarine melt rates of these glaciers, but their velocities are an order of magnitude lower than Store's, suggesting that changes in their terminus position may reflect the balance between delivery of ice to the terminus and the rate of undercutting by submarine melting. Indeed, in our melt perturbation experiments where Store did show a significant response (MD2, MC2, Figs. 3a,b, 4a,b), *maximum submarine melt rates were comparable to ice velocity*: distributed melting in MD2 varies linearly from 0 m d$^{-1}$ at the

waterline to 12 m d$^{-1}$ at the base, while conical plume melting in MC2 reaches a maximum of 24 m d$^{-1}$.

The topographic setting of Store Glacier varies greatly from north to south (Fig 1b), as does the response to environmental forcing (Fig. 3). In the north, the calving front coincides with a bedrock sill which acts as a strong basal pinning point. However, in the south, there is a bedrock sill 2.5 km behind the terminus, and this leads to flotation downstream, as the ice is

forced to accelerate over or divert around the sill. Thus, only the northern side of the terminus is well grounded, while the southern side is floating. As such, the topographic stability effect described above is weaker in the south, meaning submarine melting is most capable of influencing calving rates there, for the reasons outlined above. The stabilising effect of ice mélange is also greatest in the south, where it promotes a greater seasonal advance compared to the north. Mélange buttresses the glacier, reducing longitudinal tensile stress and inhibiting crevasse opening. In the southern floating section,

this effect stabilises the terminus, promoting advance. In the north, however, the tensile stress associated with the sudden loss of basal traction at the basal pinning point is too great to be counteracted by mélange buttressing, which means the mélange is less able to suppress calving there.

In order to explore Store's north-south divide in more detail, we plot near-terminus surface crevasse depth (Fig 6a) and basal

crevasse height (Fig 6b) predicted by the model. To investigate the role that flotation plays in crevassing and calving, we also show the 'hydrostatic imbalance' (Fig 6c), defined as:

$$\frac{-z_b}{H}\frac{\rho_{sw}}{\rho_i} - 1 \qquad\qquad\qquad (3)$$

where $\rho_i$ and $\rho_{sw}$ are the density of ice and seawater, respectively, $z_b$ is the elevation of the bed relative to sea level, and H is the thickness of the ice column. Equation 3 gives a measure of the buoyant forces acting on the ice: the value is positive

where ice is resting above neutral buoyancy, zero where ice is neutrally buoyant and negative where ice is *superbuoyant*, meaning the ice is being held *below* the level of flotation.



The most striking feature in the southern side of the terminus is the superbuoyancy downstream of the grounding line (Fig. 6c) and the associated deep (450m) basal crevasses that penetrate, in places, all the way through the glacier (Fig. 6b). This pattern arises as the ice in the south flows rapidly downhill over the southern bedrock sill (Fig 1b). The bedrock geometry and velocity of the ice mean that it continues to flow downhill beyond the level of buoyant equilibrium to become

superbuoyant. This superbuoyancy reaches a maximum halfway between the grounding line and the terminus, where surface elevation reaches a minimum of 32 m.a.s.l., beyond which point the ice begins to bend upward, as evidenced by the rising surface elevation at the terminus in the south (reaching 55 m.a.s.l.). Further evidence that this is an upward bending force comes from the complete closure of surface crevasses just beyond the grounding line (Fig 6a); this indicates compressional forces at the surface. Figure S4 shows very similar patterns during the mélange season of the control simulation, indicating

that these are persistent features relating to the broad scale geometry of the glacier.

The patterns of modelled crevasse depths and the influence of buoyancy (Fig. 6) help explain the contrasting environmental sensitivity of the northern and southern sides of Store's terminus (Fig. 3). In the north, the terminus is firmly grounded on the stoss side of a basal pinning point (Fig 1b), and there is a rapid transition from a compressive to an extensive stress regime as

the ice crests this pinning point. This is further enhanced by the sudden loss of basal drag associated with ungrounding right at the front. This transition strongly promotes calving through enhanced crevasse propagation (Benn and Evans, 2014), resulting in a topographically controlled terminus position. This explains why this side of the terminus remains comparatively fixed in position and why it displays only a limited sensitivity to the perturbations we impose. In the south, however, the upward bending associated with superbuoyancy (Fig 6c) causes deep basal crevassing up to 3 km upstream (Fig

6b). The existence of deep basal crevasses upstream of the terminus means that small perturbations to the stress regime, for example those induced by undercutting from submarine melting, can more readily promote calving than in the north. Furthermore, there is no basally controlled terminus position in the south, as the ice is has already lost contact with the bed. This more easily allows the mélange buttressing to stabilise the terminus, leading to advance in winter.

Our analysis demonstrates that basal topography and fjord geometry are critical for the stability of Greenland's marine-terminating glaciers. It also indicates that the stress conditions associated with calving are considerably more complex than previously thought; without considering 3D effects such as flotation and bending, it would not be possible to properly understand the calving process at Store Glacier and how it may respond to a warming climate. We therefore caution against over-reliance on calving models which neglect stress terms or dimensions, and emphasise the need for accurate input data for

model geometry and boundary conditions. We note, for example, that the latest IPCC report's estimates for Greenland's sea level contribution at 2100 is based on a 1D calving model, applied to only 4 outlet glaciers (Nick et al., 2013), and then upscaled on the assumption that environmental sensitivity of calving glaciers is regionally homogenous (Goelzer et al., 2013). Recent model developments by Morlighem et al. (2016) implement calving in the plan-view ice sheet model ISSM. This is a significant advance, although the need for calibration of that calving model becomes problematic once the terminus

migrates to a new position where topography is different. The results presented in this study required no calibration, which shows that 3D full-Stokes models with realistic boundary conditions and forcings can be used to predict the future behaviour of calving glaciers.

## 5. Conclusions

We have investigated glacier calving in unprecedented detail, using a novel 3D model of Store Glacier in Greenland to

resolve individual calving events. This study builds on the modelling framework developed by Todd et al. (2018) who showed that undercutting by submarine melting and buttressing by rigid proglacial melange control the glacier's seasonal





calving dynamics. Here, we demonstrate that, in the short term, the glacier is fairly insensitive to moderate changes in these processes. This most likely explains why Store's terminus has remained stable over the past four decades, including a recent period when many other glaciers in the Uummannaq region, and elsewhere in Greenland, underwent concurrent and sustained retreat (Howat et al. 2010, Seale et al. 2011). However, we found that more severe changes associated with a

warming climate probably will undermine this stability. A complete loss of buttressing due to disappearance of the ice mélange may prevent terminus advance in winter while exacerbating summer retreat. A doubling of submarine melt rates, as either distributed or concentrated plumes or both, may lead to retreat beyond the current observed range. Behind the current terminus position, Store's basal topography becomes much deeper (Fig. 1b); if the terminus were to retreat to this point, this might trigger sustained interannual retreat.

An important outcome of this study is that neither the calving style nor the environmental sensitivity of calving glaciers is spatially uniform. For Store Glacier, calving in the south side is driven by deep basal crevassing and is significantly modified by both melt undercutting and mélange buttressing. In the north, calving is driven by surface crevassing and is comparatively insensitive to external forcing. While topography clearly exerts a strong influence on the glacier's calving behaviour, only

half of the terminus is topographically pinned, and this partial stability may render the glacier more prone to retreat and less stable compared to recent work (Catania et al., 2018). Remote sensing studies (Carr et al., 2013; McFadden et al., 2011; Moon et al., 2015) suggest that topography may be an important control on calving at other West Greenland outlet glaciers. The importance of topographic control lends strong support to the use of fracture criteria for predicting calving. Our results suggest that, for glaciers with topographically well-defined terminus positions, calving occurs at specific *places* rather than

at specific *rates*. The ongoing search for universal calving laws should incorporate improved prediction of fracture processes, rather than simply linking hypothetical calving rates to environmental or dynamic factors. Our results also strongly support the use of 3D full-Stokes models in calving research. Flowline or vertically-integrated models will fail to capture lateral variability and vertical stress gradients, respectively, both of which appear to be important at Store Glacier. Finally, our results indicate the importance of accurate input data, particularly basal topography. In this regard, the

BedMachine product (Morlighem et al., 2017) has proved to be a vital step forward in Greenland calving research.

### Code availability

The code constituting the 3D calving model has been made freely available via integration with the Elmer/Ice source code, available at: https://github.com/ElmerCSC/elmerfem/tree/elmerice.

### Author Contribution

JT developed the calving model with substantial technical assistance from TZ and PR and scientific guidance from PC and DB. JT and PC designed the sensitivity experiments and interpreted the results with assistance from DB and TZ. JT prepared the manuscript with contributions from all co-authors.

### Acknowledgements

This study was funded by the Natural Environment Research Council through a PhD studentship to JT (grant no.

NE/K500884/1) and a research grant (NE/K005871/1) to PC. We acknowledge that the results of this research have been achieved using the PRACE-3IP project DynaMITE (FP7 RI-312763) awarded to JT and PC with resource Sisu based in Finland at CSC. PC also acknowledges support from the European Research Council under the European Union's Horizon 2020 Research and Innovation programme (grant agreement 683043). We thank Michiel van den Broeke for providing





RACMO climate data, and the Elmer/Ice developer community for their support. Landsat Surface Reflectance products courtesy of the U.S. Geological Survey.

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




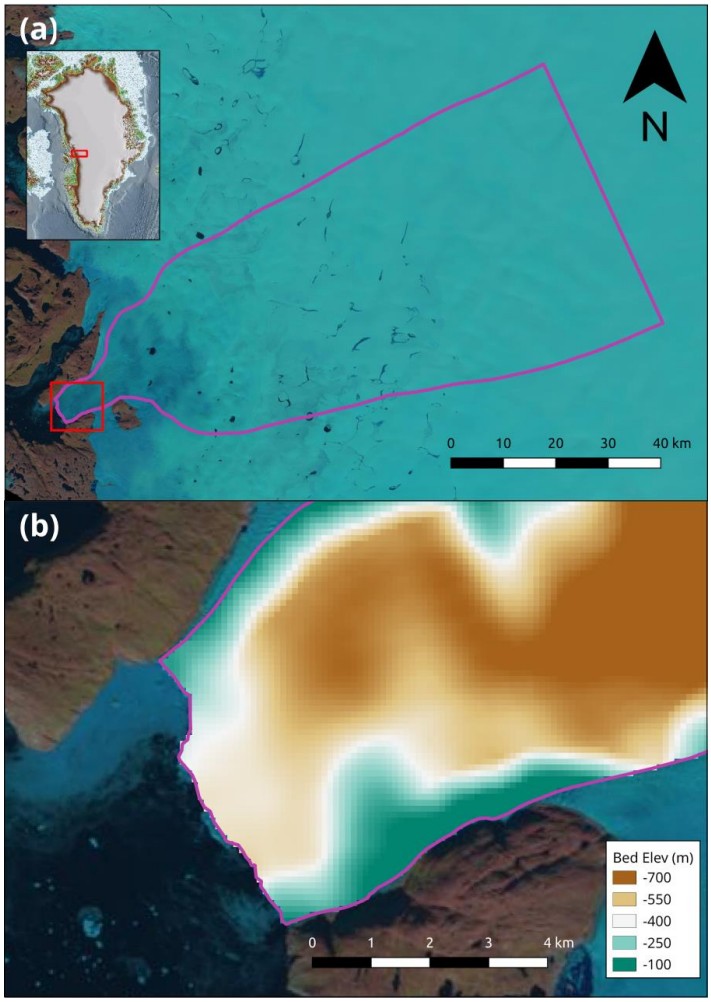

**Figure 1: (a) Site map showing modelled domain of Store Glacier. (b) Basal topography near the glacier terminus.**





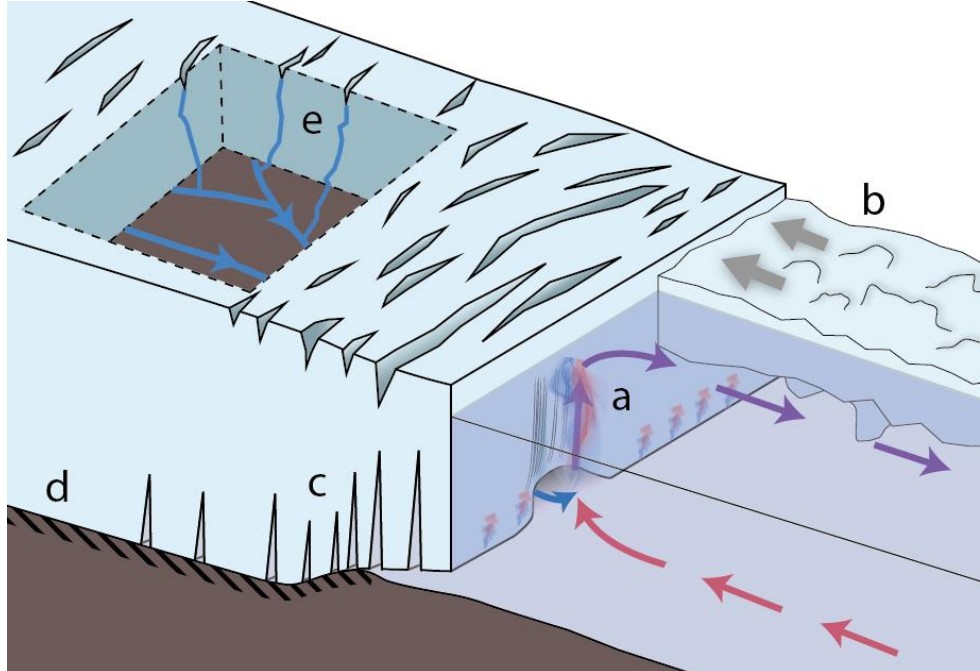

**Figure 2: Processes affecting calving of Greenland's marine-terminating glaciers. a) Distributed and concentrated plumes accelerate melting which undercut the ice front. This promotes extensional stress which opens crevasses at the surface of the glacier. b) Ice mélange provides buttressing which closes crevasses and supports the terminus. c) Basal pinning points and lateral constrictions inhibit crevasse penetration in the stoss side and enhance it in the lee side d) Spatial and temporal changes in basal drag affect stress patterns and crevassing. e) The basal hydrological system discharge cold and fresh, glacial meltwater, which feed the plumes that cause high rates of submarine melting.**

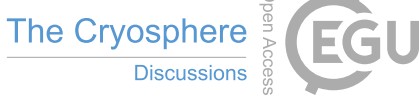

| Run Code | Ice Mélange Thickness (m) | Submarine Melt | | |
|---|---|---|---|---|
| | | Dist. Summer Ave. (m d$^{-1}$) | Dist. Winter Ave. (m d$^{-1}$) | Conc. Max (m d$^{-1}$) |
| CONTROL | 140 | 3.1 | 1.3 | 12 |
| MM1 | 70 | 3.1 | 1.3 | 12 |
| MM2 | 0 | 3.1 | 1.3 | 12 |
| MD1 | 140 | 4.65 | 1.95 | 12 |
| MD2 | 140 | 6.2 | 2.6 | 12 |
| MC1 | 140 | 3.1 | 1.3 | 18 |
| MC2 | 140 | 3.1 | 1.3 | 24 |
| MA1 | 70 | 4.65 | 1.95 | 18 |
| MA2 | 0 | 6.2 | 2.6 | 24 |

**Table 1: Simulations performed in the sensitivity analysis. Run codes begin 'M' to distinguish (M)agnitude perturbations as opposed to (D)uration perturbations (Text S1). Two simulations each were performed for (M)élange, (D)istributed melt, (C)oncentrated melt and (A)ll forcings. The digit refers to either the 1st or 2nd perturbation, where the 2nd is more 'severe'. Green highlights the parameter values which deviate from the control simulation (i.e. the active perturbation).**



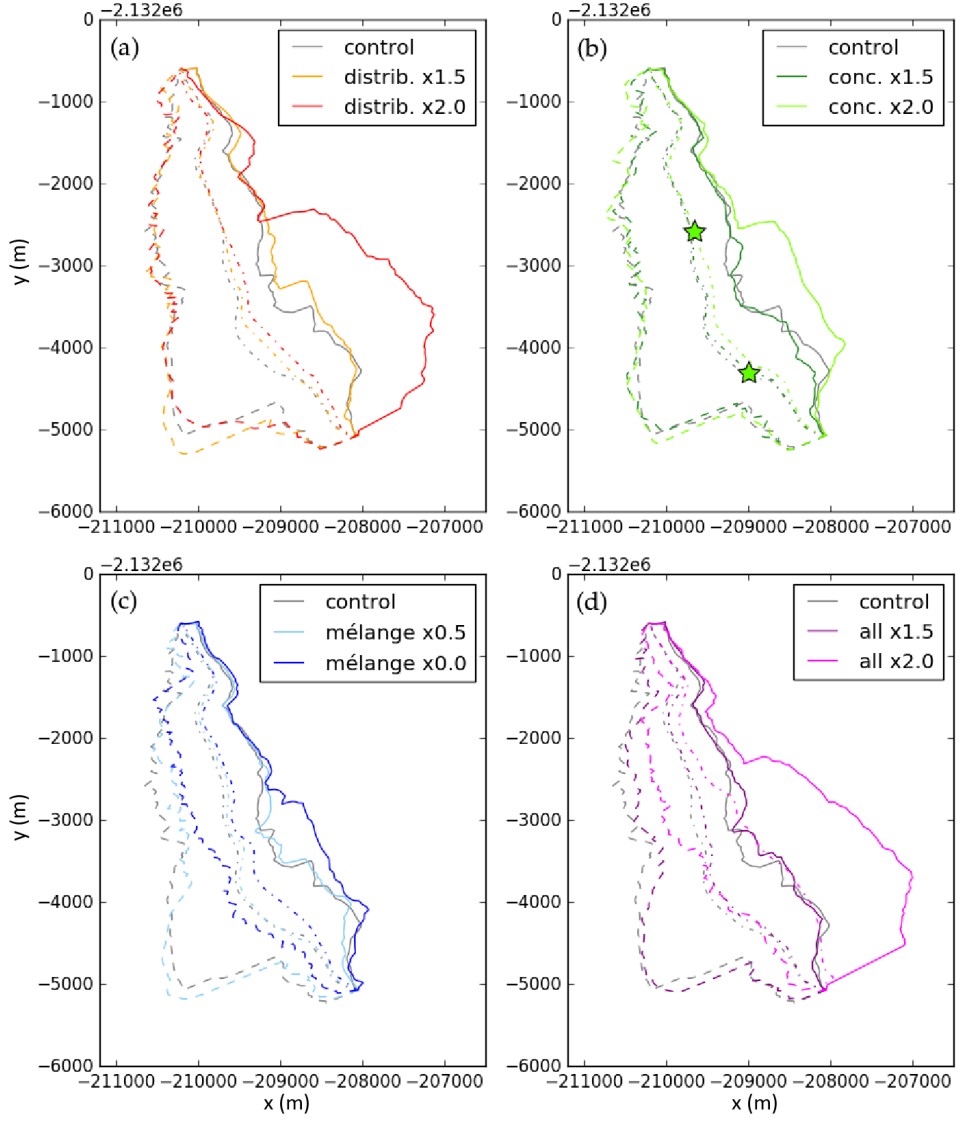

**Figure 3: Maximum (dashed), mean (dash-dotted) and minimum (solid) terminus positions for step changes in a) distributed melt rate (MD1,2), b) concentrated melt rate (MC1,2), c) mélange thickness (MM1,2), and d) all of the above (MA1,2). Green stars in (b) show the position of the two concentrated plumes. Grey lines indicate the control simulation, and increasingly bright colour indicates more severe environmental forcing.**





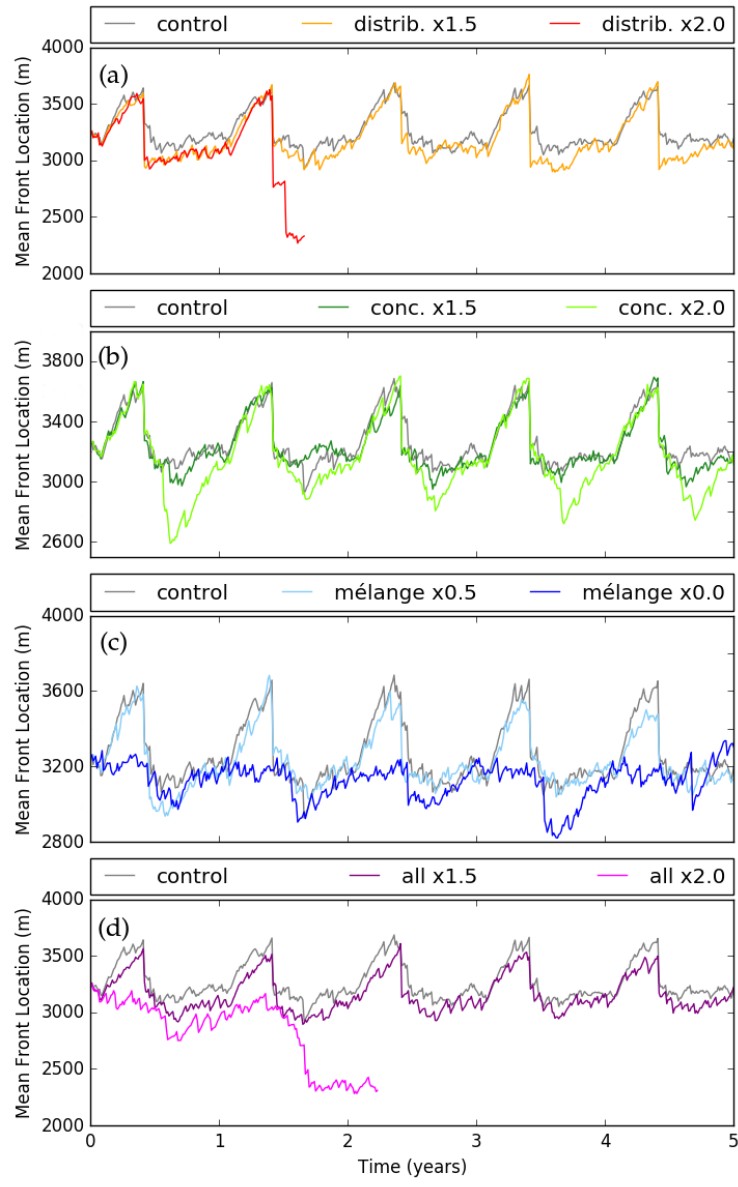

Figure 4: Mean terminus position over 5 years for step changes in a) distributed melt rate (MD1,2), b) concentrated melt rate (MC1,2), c) mélange thickness (MM1,2), and d) all of the above (MA1,2). Note different y-axis scales.



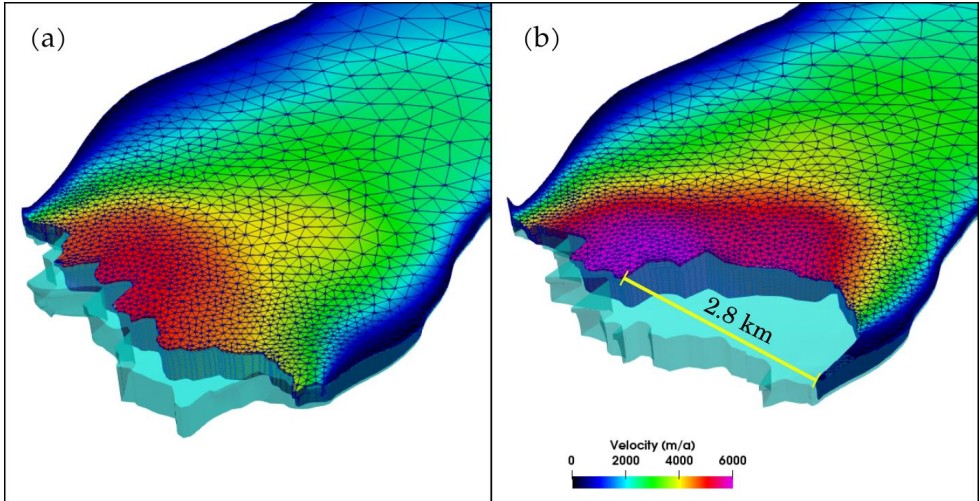

**Figure 5: (a) Typical seasonal range in terminus position for control simulation (present day forcing). (b) Collapse of the southern side of the terminus in Run MD2 (distributed melting x 2) prior to simulation break down.**

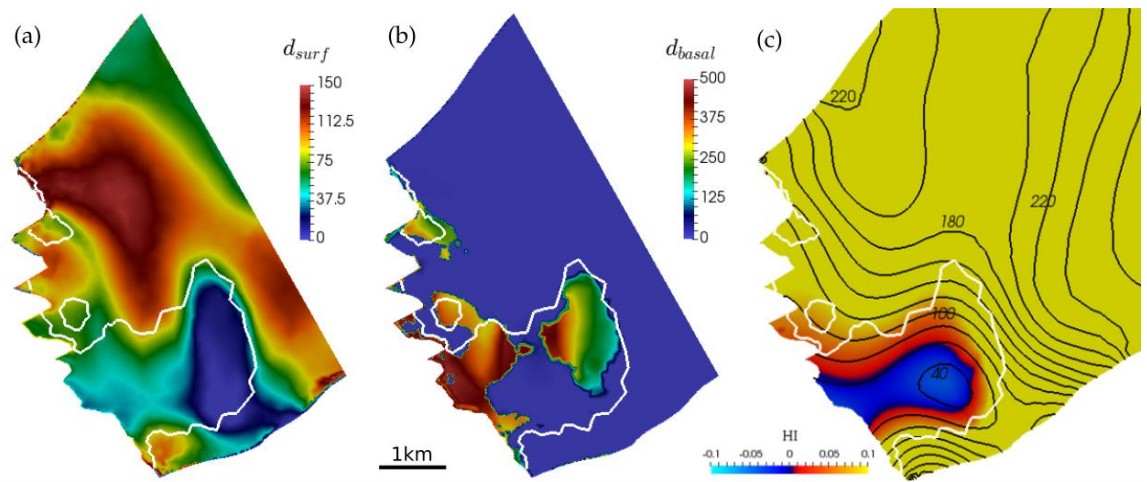

**Figure 6: (a) Surface crevasse penetration and (b) basal crevasse penetration from the beginning of the control simulation. (note different colour scale.) (c) Hydrostatic imbalance (Eq. 1) at the terminus and 20m surface elevation contours (black lines). White line indicates the grounding line. Figure S4 shows the equivalent panels for the mélange season, indicating that these patterns are permanent features of the glacier.**

