# Peer review of "Sensitivity of a calving glacier to ice-ocean interactions under climate change: New insights from a 3D full-Stokes model"

_The Cryosphere, 2019_

## Referee Comment (RC1) · Anonymous Referee #1 · 19 Mar 2019

*Review of 'Sensitivity of calving glaciers to ice-ocean interactions under climate change: New insights from a 3D full-Stokes model'*

**General comments**

Using a 3D full-Stokes calving model, the authors explore the sensitivity of Store Glacier in western Greenland under changing magnitudes and durations of submarine melt and mélange buttressing. Scientific quality of the work is high. Calving is an important and active area of research at the moment, and this paper will be a welcome additional reference for further modelling and observational research. The paper is well-written, with appropriate figures, and provides an interesting addition to understanding how best to model calving processes, as well as further insight into processes at Store Glacier. Methods are well explained and the discussion flows nicely from the presented results. The work is also placed in context with other modeling studies, including specifically of Store Glacier. Some additional attention to clear and consistent use of specific language is needed, as well as additional (but brief) points of explanation. These are noted in the specific comments below.

**Specific comments (by page/line number, including technical corrections)**

Title. Since this study only examines a single glacier, the first half of the title is a bit of a reach. I'm OK with keeping it as is, but the editor may want to consider suggesting a change.

1/10. Not sure it's accurate that calving rates/processes is 'one of the largest uncertainties'. Perhaps just 'a substantial uncertainty'.

1/12. I would not normally consider that there is one calving mechanism. Suggest changing 'the calving mechanims' to 'calving mechanisms'

1/16. Replace 'equivalently' with 'by 50%'

1/21. Specify if referring to surface or subglacial melt rate

1/25. Replace 'the most' with 'an'

1/32-33. Sustained acceleration does not always result from increased calving. Also, 'environmental forcings' is vague. Revise sentence to be more specific about forcings and note that sustained acceleration is not always a result of increased calving.

2/6. Recommend replacing 'routed along the bed' to 'discharged at the terminus-bed interface', because the discharge is playing the important role in this case, not the subglacial system.

2/14. This introductory sentence specifies that *internal* dynamics strongly modulate the effect of external forcing on calving, but subsequent sentences focus on external factors (topography, basal friction). The argument that internal dynamics module calving effect needs strengthening if that's the main point.

2/19. Change 'potentially' to 'potential'

3/8+. This section does not properly distinguish between distributed and plume-induced (contracted) melt. Begin to use the desired language right away and maintain clarity.

3/16. Any sense of how representative these temperature-salinity records are over multiple years? Expectations for future changes? (Could also go into discussion).

3/32. Change to 'third environmental forcing we consider is…'

3/34. Add citation of Amundson et al. 2010 (already in references)

3/38. Add a sentence summarizing the observations of 140 m thickness. Seems surprisingly thick and nice to have a quick explanation instead of sending the reader to another paper.

4/7-9. Please provide a bit more detail regarding the difficulty with scaling so that it's clearer why you applied a different method.

4/16. It's important to explain why the simulation broke down at the first mention of this problem. Move this information from the bottom of page 5.

5/17. Remove 'greatly'

5/30. Why did rapid change not sent in until year 2?

7/1-2. Change 'firstly' and 'secondly' to 'first' and 'second'

7/5+. Add another sentence or two on what was tested re: duration changes. 'Duration' can cover a wide variety of differences with one end member being the same as completely eliminating seasonal changes. Best to quickly sum up the experiments and results, while details remain in the supplementary text.

7/8-9 and 8/41-42. Reconsider the language used re: 'intermediate scenarios' and 'climate change becomes severe'/'severe warming'. These are much too vague. There is a fairly specific science community notion of what an intermediate scenario is (IPCC), while 'severe' has little to no quantitative meaning. If you want only to refer to specific 'scenarios' used in this paper, then you need to be very clear in consistently calling them 'scenarios' and can assign individual runs to 'intermediate', 'severe', etc. For example, do you want to define 'intermediate' at 50% more submarine melt and/or 50% less buttressing, as suggested in the abstract, or in some other way? Again, this must be laid out clearly and used throughout the paper/discussion.

7/13. Are there no other datasets that can help with speculation on continued retreat? For example, BedMachine v3. There's no reason that speculation must only be based on data within this study. Consider looking to other science resources.

7/19+. I was surprised that surface thinning is not discussed. Surface thinning certainly plays into the processed discussed in the paper, and should be at minimum briefly mentioned somewhere in the discussion.

7/36. Again, mention the magnitude of change in duration examined.

10/5+. Are these modeled or observed? Make sure to always be clear about this distinction. Also, do the comments regarding surface character from the model align with imagery?

10/35-37. 'can be used to predict the future behavior' may be too strong a statement. Additional comparisons between model and observed behavior are warranted before moving to stronger statements like this.

11/4+. Recommend commenting on how realistic any of these changes are. For example, the loss of mélange seems highly unlikely in cases where calving rates remain at current levels because mélange is strongly controlled by iceberg production.

11/25. Suggest noting need for continued improvement of bed topography data, especially near the terminus. There are still some substantial errors in BedMachine v3.

Figure 1. What does the velocity field look like for this region? Is there any lateral input from the south that is excluded from the model domain?

Figure 2. Does mélange buttressing really *close* crevasses, or simply suppresses calving? Consider change. Correct last sentence to read 'system discharges cold and fresh glacial meltwater, which feeds the'. Consider further exaggerating the difference in basal crevasses on the stoss/less sides in the graphic.

Figure 3. Remove number labels at top left of each plot and add MD1/2, MM1/2 labels. Can you add a background image of the glacier to help the reader visualize the scale of change (Landsat image would work well)?

Figure 4. Clarify the direction of advance v. retreat. Clarify if the tick mark of each year is meant to be January. Mention in caption why the lines in a) and d) are cutoff (stimulation breakdown). While I understand that you may not want to make all y-axis scales the same, matching as many as possible is helpful – a) and d) already the same, consider making b) and c) the same.

Figure 6. What terminus front position is shown?

Supplementary text (by page)

Table S1. Best to use full words in the column labels as much as possible. Using 'mean' instead of 'average' can help with space, while 'distributed' and 'concentrated' are more difficult (give it a moment of creative thought in case you can make it work).

Page 2. At the top, you say that combining magnitude and duration perturbations significantly changes terminus behavior. After reading the main text, in which you state that duration experiments didn't change glacier behavior much, I was not expecting to find 'significant changes' resulting from anything contained within the supplementary text. Reconsider the content of the main and supplementary text to avoid this surprise.

Page 2. First sentence of last paragraph – Clarify whether DM2 is the 'most aggressive' perturbation of all experiments or just the ones discussed in the supplementary text.

Figure S2. Make y-axis for b) and d) 4000-6000.

---

## Referee Comment (RC2) · Mathieu Morlighem (Referee) · 21 Mar 2019

The paper "Sensitivity of calving glaciers to ice-ocean interactions under climate change: New insights from a 3D full-Stokes model" by Joe Todd, Poul Christoffersen, Thomas Zwinger, Peter Råback, and Douglas Benn investigates the response of Store Glacier's ice front position to various forcings using a high-resolution full Stokes model. They examine the effect of an increase in undercutting by submarine melting, the effect of a concentrated vs distributed melt rates at the calving face, and a reduction in the backstress exerted by ice mélange. Overall, the authors find that Store is stable for a wide range of conditions, but starts to retreat dramatically for the strongest scenarios.

[Figure]

The authors also highlight the important role of bed geometry in ice front dynamics.

The paper is well written and easy to follow. Calving is a critically important process that needs to be better understood in order to reduce the uncertainty of projections from ice sheet models. This is an important and timely contribution that I recommend for publication in TC after some minor revisions. I have some suggestions that I hope will help improve the manuscript.

**1 General Comments**

I understand that the model setup is described in Todd et al. 2018, but I still think that some important model characteristics, such as mesh resolution, time stepping, or the undercutting parameterizations should be mentioned again. More importantly, one key aspect of this model compared to other existing models is that (if I understand correctly) the calving face *is not* assumed to be vertical. The effect of undercutting is therefore more rigorously simulated, which is a significant advantage over other models. This should be clearly stated in the text, as most models (including mine!) model undercutting as an extra calving rate as the calving face is always vertical.

Another important point is that some physical processes discussed in this paper are assumed to be fully established and validated, while they are actually still poorly understood or controversial. The effect of ice mélange for example: it has been proposed that ice mélange could prevent iceberg from overturning, thereby inhibiting calving. This has been shown by Amundsen et al. 2010. However, the jury is still out when it comes to the potential buttressing effect (i.e. mechanical stress) that ice mélange would exert on the ice stream. Some other studies suggest that the presence of sea ice is only a reflection of oceanic conditions that are actually the dominating control on calving. While it seems clear that ice mélange may prevent calving, it is still not clear whether ice mélange has any (direct) effect in terms of buttressing.

Another important process is how crevasses propagate. The Nye approximation remains a simple approximation and has not (to my knowledge) been fully validated. The presence of basal crevasses upstream of grounding line is not seen in most of the radar echograms that I have seen. Basal crevasses start to form at the grounding line where tidal bending occurs, but (to my knowledge) are not visible under grounded ice. The authors present the calving law used here as "the truth", but it remains one possible (promising) representation of calving. I think the text should be less definitive in some places.

I obviously also have to say a few words about the comparison with Morlighem et al. 2016. The study presented here is undeniably more sophisticated but I still think that the authors have not rigorously demonstrated that a depth integrated model would respond differently. It is very difficult to compare the model from Morlighem et al. 2016 and the one presented here: they have a different initial state (this one is relaxed, so it may start from a different surface height), we use different meshes, possibly a different bed, different boundary conditions (I did not model lateral friction), etc. For example, I did not account for the presence of mélange: the inversion of basal drag is therefore expected to yield a slightly higher stress compared to the one of this model since the ice front includes here a stronger back stress from ice mélange. This will have an impact on the model sensitivity: my model having a stronger basal stress will be more stable. While both studies agree on many points (e.g., overall stability of Store, strong control of the bed geometry, etc), it is very difficult to disentangle why the models require different melt rates to be dislodged from their current position. We cannot claim that the model presented here is "better" simply because it is based on Full-Stokes. It is not what is shown. The only way to show that would be to collapse the model and run it with an SSA approximation (Elmer has this capability): that would make it possible to compare apples with apples. It would be great if this could be tested, but if this is not possible, the text needs to be less definitive in places (e.g., use conditional instead of present tense). ISSM also has a full-Stokes solver and from my experience, the results have almost always been very similar to results obtained with SSA. That being said,

this was for vertical calving fronts.

Finally, I found a bit unfortunate that the model breaks as soon as something interesting happens. I am surprised that reducing the time step does not fix the problem. Is there any possible way to improve the stability of the numerical implementation? At the end of the day, we want to be able to model ice front retreat, not just demonstrate its stability...

Again, I think this is a great piece of work, I just think that a few things need to be put in perspective and some statements need to be nuanced.

**2 Specific comments**

- title: the title is too generic. This paper is about one specific glacier, Store, I am not sure why the title says "Sensitivity of calving glaciers", it should be "Sensitivity of Store Glacier to ..."

- p1 l38: Figure 2 is referenced after figure 1 (only mentioned page 2 line 26), maybe the order of the figures should be changed

- p3 l24: 162 m (space between number of m, also l37 and in other places)

- p7 l20: this statement is misleading, ISSM is not "depth-averaged". We implemented several ice flow models in ISSM, including SSA and full-Stokes. In Morlighem et al. 2016, the depth-averaged SSA model was using, but ISSM itself is not depth-averaged.

- p9 l9: -400 m a.s.l. sounds a bit awkward, maybe replace by 400 below sea level?

- p9 eq3: the hydrostatic imbalance quantity is difficult to appreciate. I think using the height above floatation (multiplying this quantity by $H\,\rho_i$) would make it easier

to evaluate how much the ice has to thin to be floating. The figure should also be adjusted accordingly.

- p19 figure 3: all subplots have -2.132e6 at the top of the y axis, it is probably the offset for $y$, but it is not clear... It is a bit confusing, I am not sure it is necessary.

---

## Author Comment (AC1) · 16 May 2019

**Repsonse to Reviewer Comment by Anonymous Reviewer on "Sensitivity of calving glaciers to ice-ocean interactions under climate change: New insights from a 3D full-Stokes model" by Todd et al.**

We are very grateful to the anonymous reviewer for their feedback on this manuscript. We provide responses to reviewer comments in red below. Page and line numbers refer to the updated manuscript.

**General comments**

Using a 3D full-Stokes calving model, the authors explore the sensitivity of Store Glacier in western Greenland under changing magnitudes and durations of submarine melt and mélange buttressing. Scientific quality of the work is high. Calving is an important and active area of research at the moment, and this paper will be a welcome additional reference for further modelling and observational research. The paper is well-written, with appropriate figures, and provides an interesting addition to understanding how best to model calving processes, as well as further insight into processes at Store Glacier. Methods are well explained and the discussion flows nicely from the presented results. The work is also placed in context with other modeling studies, including specifically of Store Glacier. Some additional attention to clear and consistent use of specific language is needed, as well as additional (but brief) points of explanation. These are noted in the specific comments below.

**Specific comments** (by page/line number, including technical corrections)

Title. Since this study only examines a single glacier, the first half of the title is a bit of a reach. I'm OK with keeping it as is, but the editor may want to consider suggesting a change.

We have changed the title of the article to "Sensitivity of a calving glacier to ice-ocean interactions..."

1/10. Not sure it's accurate that calving rates/processes is 'one of the largest uncertainties'. Perhaps just 'a substantial uncertainty'.

OK, changed accordingly.

1/12. I would not normally consider that there is one calving mechanism. Suggest changing 'the calving mechanims' to 'calving mechanisms'

Done

1/16. Replace 'equivalently' with 'by 50%'

Done

1/21. Specify if referring to surface or subglacial melt rate

Good point – done.

1/25. Replace 'the most' with 'an'

Done

1/32-33. Sustained acceleration does not always result from increased calving. Also, 'environmental forcings' is vague. Revise sentence to be more specific about forcings and note that sustained acceleration is not always a result of increased calving.

We have changed this sentence to:

"The importance of calving as a contributor to global sea level rise is demonstrated by the sustained retreat, acceleration and dynamic thinning triggered at the termini of many Greenlandic outlet glaciers in the past two decades."

We agree that sustained acceleration does not always result from increased calving, but we believe the pattern is sufficiently common to warrant a broad statement to that effect.

2/6. Recommend replacing 'routed along the bed' to 'dischwarged at the terminus-bed interface', because the discharge is playing the important role in this case, not the subglacial system.

We investigate the different responses of the glacier to concentrated & distributed submarine melting, and so the nature of the subglacial hydrological system is likely critical. We have added "to the terminus" to capture both aspects.

2/14. This introductory sentence specifies that internal dynamics strongly modulate the effect of external forcing on calving, but subsequent sentences focus on external factors (topography, basal friction). The argument that internal dynamics module calving effect needs strengthening if that's the main point.

Basal friction is certainly modified, in part, by external forcing, but these factors (topography, geometry & drag) effectively define the glacier's internal dynamics. We believe this is a question of semantics – we have removed the 'internal' to avoid confusion.

2/19. Change 'potentially' to 'potential'

Done

3/8+. This section does not properly distinguish between distributed and plume-induced (contracted) melt. Begin to use the desired language right away and maintain clarity.

We only consider forced convective (i.e. plume-induced) melting in this study, due to the dominance of this effect at Store Glacier. We decompose the observed complex plume pattern into a background 'distributed/planar' plume and 'concentrated/conical' plumes. For clarity, we now introduce the concepts of "*distributed ('planar') plume melting*" and "*concentrated ('conical') plume melting*" [P3L11], then refer consistently to concentrated and distributed plume melting throughout the text.

3/16. Any sense of how representative these temperature-salinity records are over multiple years? Expectations for future changes? (Could also go into discussion).

We have added a sentence to the discussion [P9L12] to state that "any future changes to the stratification of the fjord water in front of Store glacier could affect buoyant plume behaviour and melt rates".

3/32. Change to 'third environmental forcing we consider is...'

Good point – done.

3/34. Add citation of Amundson et al. 2010 (already in references)

Done

3/38. Add a sentence summarizing the observations of 140 m thickness. Seems surprisingly thick and nice to have a quick explanation instead of sending the reader to another paper.

We have modified this sentence to clarify that 140m thickness estimates come from UAV survey in 2014: "Mélange thickness is based directly on observations from photogrammetric UAV survey in 2014 (Toberg et al., 2016, Todd et al., 2018)…" [P4L6]

4/7-9. Please provide a bit more detail regarding the difficulty with scaling so that it's clearer why you applied a different method.

We have added "melt rates scale sublinearly with subglacial discharge (Xu et al., 2013), meaning that disproportional amounts of additional discharge were needed to double the maximum melt rate" [P4L17]

4/16. It's important to explain why the simulation broke down at the first mention of this problem. Move this information from the bottom of page 5.

Done – we have also expanded the explanation slightly and flagged this as a priority for future model development.

5/17. Remove 'greatly'

Done

5/30. Why did rapid change not sent in until year 2?

The effect of the increased melt on the terminus mass budget in year 1, while insufficient to destabilise the terminus, preconditions the glacier for collapse in year 2. We have added a sentence in the discussion: "For instance, in Run MD2, although we force the model with doubled melt rates in summer from the beginning, broad-scale terminus retreat does not begin until year 2, indicating the importance of the interannual mass budget." [P8L22]

7/1-2. Change 'firstly' and 'secondly' to 'first' and 'second'

Done

7/5+. Add another sentence or two on what was tested re: duration changes. 'Duration' can cover a wide variety of differences with one end member being the same as completely eliminating seasonal changes. Best to quickly sum up the experiments and results, while details remain in the supplementary text.

Good point – we have clarified this with a brief summary [P7L26].

7/8-9 and 8/41-42. Reconsider the language used re: 'intermediate scenarios' and 'climate change becomes severe'/'severe warming'. These are much too vague. There is a fairly specific science community notion of what an intermediate scenario is (IPCC), while 'severe' has little to no

quantitative meaning. If you want only to refer to specific 'scenarios' used in this paper, then you need to be very clear in consistently calling them 'scenarios' and can assign individual runs to 'intermediate', 'severe', etc. For example, do you want to define 'intermediate' at 50% more submarine melt and/or 50% less buttressing, as suggested in the abstract, or in some other way? Again, this must be laid out clearly and used throughout the paper/discussion.

Thanks – this is a good point. We have modified the text to avoid referring to 'intermediate or severe' *climate warming scenarios* – we agree that this is potentially confusing/misleading. On [P7L34] we now state:

"Our results indicate that Store Glacier should remain relatively unchanged under intermediate perturbation scenarios (MD1, MC1, MM1, MA1) , but that it may undergo retreat in response to more severe forcing (MD2, MC2, MM2, MA2)."

and on [P9L37] we now have:

"The results indicate that overall, Store is relatively insensitive to moderate changes in terminus forcing associated with climate warming. Under more severe perturbation, the model shows…"

We hope that this clarifies to the reader that we are referring specifically to our intermediate and extreme simulations.

7/13. Are there no other datasets that can help with speculation on continued retreat? For example, BedMachine v3. There's no reason that speculation must only be based on data within this study. Consider looking to other science resources.

In the conclusions [P12L11] we state, with reference to Fig 2b, that the inland overdeepening implies that any initial retreat may be sustained.

7/19+. I was surprised that surface thinning is not discussed. Surface thinning certainly plays into the processed discussed in the paper, and should be at minimum briefly mentioned somewhere in the discussion.

We have added the following sentence to the discussion [P8L23]: "Sustained interannual mass loss near the terminus could lead to dynamic thinning; the simulations performed here are not long enough to capture this effect, but the positive feedback between retreat, acceleration and thinning could be a major destabilising influence."

7/36. Again, mention the magnitude of change in duration examined.

We have added: ", even when the summer season is extended by 2 months,"

10/5+. Are these modeled or observed? Make sure to always be clear about this distinction. Also, do the comments regarding surface character from the model align with imagery?

We have added the clarifier 'modelled' at [P10L31] and [P10L40]. We have also added a sentence stating, with reference to Todd et al., (2018) that the modelled & observed surface are in good agreement [P11L7].

10/35-37. 'can be used to predict the future behavior' may be too strong a statement. Additional comparisons between model and observed behavior are warranted before moving to stronger statements like this.

Changed to [P11L35]: "which suggests that 3D full-Stokes models, with realistic boundary conditions and forcings, have the potential to shed new light on the future behaviour of calving glaciers."

11/4+. Recommend commenting on how realistic any of these changes are. For example, the loss of mélange seems highly unlikely in cases where calving rates remain at current levels because mélange is strongly controlled by iceberg production.

We believe that not enough is known about the processes of mélange formation to speculate about the likelihood of its complete disappearance. It may be that persistent sea ice and/or cold ocean/air temperatures are a prerequisite for formation. Furthermore, the iceberg size distribution may be critical, and this may change as the glacier evolves. We have added a sentence at [P9L31]:

"Suppression of calving by mélange occurs primarily during winter and spring, when iceberg dispersal is prevented by sea ice. The effectiveness of buttressing can be greatly reduced if rigid mélange fails to form due to warmer air temperatures or surface water, as appears to have happened at Kangerdlugssuaq in 2017-18 (Bevan et al., 2019). Loss of mélange buttressing as implemented in Runs MM2 & MA2 is therefore possible in a warming world."

With regards to the feasibility of a doubling of submarine melting, we have commented on this in the discussion [P9L1].

We have changed "more severe changes… *will* undermine this stability" to "*would* undermine this stability" to avoid implying that these severe perturbations will necessarily occur.

11/25. Suggest noting need for continued improvement of bed topography data, especially near the terminus. There are still some substantial errors in BedMachine v3.

While we agree that there is some work to do in this area, we feel that to address this here makes for a rather confusing conclusion, so we have left it as is.

Figure 1. What does the velocity field look like for this region? Is there any lateral input from the south that is excluded from the model domain?

We have added an overlay of velocity from MEaSUREs data to panel (a). The lateral flux from the southern tributary is insignificant due to 1) the low velocity and 2) the low thickness.

Figure 2. Does mélange buttressing really close crevasses, or simply suppresses calving? Consider change. Correct last sentence to read 'system discharges cold and fresh glacial meltwater, which feeds the'. Consider further exaggerating the difference in basal crevasses on the stoss/less sides in the graphic.

We have changed this to "Ice mélange provides buttressing which supports the terminus and suppresses calving", and fixed the grammatical errors in point e) – Thanks for spotting. We have made a similar modification to the text at [P9L21].We have also exaggerated the basal crevasse propagation in Fig 2 (now Fig 1) as suggested.

Figure 3. Remove number labels at top left of each plot and add MD1/2, MM1/2 labels. Can you add a background image of the glacier to help the reader visualize the scale of change (Landsat image would work well)?

We've totally overhauled this figure and agree that it now looks much better – thanks for the suggestions.

Figure 4. Clarify the direction of advance v. retreat. Clarify if the tick mark of each year is meant to be January. Mention in caption why the lines in a) and d) are cutoff (stimulation breakdown). While I understand that you may not want to make all y-axis scales the same, matching as many as possible is helpful – a) and d) already the same, consider making b) and c) the same.

Thanks for these helpful suggestions - done.

Figure 6. What terminus front position is shown?

This is from the beginning of the control simulation, as stated in the caption.

Supplementary text (by page)

Table S1. Best to use full words in the column labels as much as possible. Using 'mean' instead of 'average' can help with space, while 'distributed' and 'concentrated' are more difficult (give it a moment of creative thought in case you can make it work).

We've changed 'ave.' to 'mean' as suggested, but struggled to find a way to expand 'distributed' and 'concentrated' without lowering the font size, which negatively affects the overall readability. We have changed 'Dist.' to 'Distrib.', which will hopefully be more easily understood.

Page 2. At the top, you say that combining magnitude and duration perturbations significantly changes terminus behavior. After reading the main text, in which you state that duration experiments didn't change glacier behavior much, I was not expecting to find 'significant changes' resulting from anything contained within the supplementary text. Reconsider the content of the main and supplementary text to avoid this surprise.

This is a significant change compared to the 'duration only' experiments. The results are actually very similar to the 'magnitude only' experiments presented in the main text. We should have been clearer about this. We have modified this sentence to read:

"Compared with the 'duration' experiments, combining both magnitude and duration perturbations significantly changes the terminus behaviour; the glacier responds in a manner which is qualitatively similar to the 'magnitude' experiments discussed in the main text."

Page 2. First sentence of last paragraph – Clarify whether DM2 is the 'most aggressive' perturbation of all experiments or just the ones discussed in the supplementary text.

Good point. Appended 'in this study'.

Figure S2. Make y-axis for b) and d) 4000-6000.

Done – also set same Y axis scale for panels (a) & (c).

---

## Author Comment (AC2) · 16 May 2019

**Repsonse to Reviewer Comments by Matthieu Morlighem on "Sensitivity of calving glaciers to ice-ocean interactions under climate change: New insights from a 3D full-Stokes model" by Todd et al.**

We are very grateful to Matthieu Morlighem for his feedback on this manuscript. We provide responses to reviewer comments in red below. Page and line numbers refer to the updated manuscript.

The paper "Sensitivity of calving glaciers to ice-ocean interactions under climate change: New insights from a 3D full-Stokes model" by Joe Todd, Poul Christoffersen, Thomas Zwinger, Peter Råback, and Douglas Benn investigates the response of Store Glacier's ice front position to various forcings using a high-resolution full Stokes model. They examine the effect of an increase in undercutting by submarine melting, the effect of a concentrated vs distributed melt rates at the calving face, and a reduction in the backstress exerted by ice mélange. Overall, the authors find that Store is stable for a wide range of conditions, but starts to retreat dramatically for the strongest scenarios.

The authors also highlight the important role of bed geometry in ice front dynamics. The paper is well written and easy to follow. Calving is a critically important process that needs to be better understood in order to reduce the uncertainty of projections from ice sheet models. This is an important and timely contribution that I recommend for publication in TC after some minor revisions. I have some suggestions that I hope will help improve the manuscript.

**1 General Comments**

I understand that the model setup is described in Todd et al. 2018, but I still think that some important model characteristics, such as mesh resolution, time stepping, or the undercutting parameterizations should be mentioned again. More importantly, one key aspect of this model compared to other existing models is that (if I understand correctly) the calving face is not assumed to be vertical. The effect of undercutting is therefore more rigorously simulated, which is a significant advantage over other models. This should be clearly stated in the text, as most models (including mine!) model undercutting as an extra calving rate as the calving face is always vertical.

Good point. We have added a sentence describing the mesh resolution (P5L8), a couple of sentences briefly describing the timestepping and pointing the reader to Todd et al. (2018) for additional details (P5L18), and a more detailed description of our plume melting implementation (P3L15). We have also added a paragraph to Section 2.4 (P5L21) to highlight the fact that the calving front can be non-vertical.

Another important point is that some physical processes discussed in this paper are assumed to be fully established and validated, while they are actually still poorly understood or controversial. The effect of ice mélange for example: it has been proposed that ice mélange could prevent iceberg from overturning, thereby inhibiting calving. This has been shown by Amundsen et al. 2010. However, the jury is still out when it comes to the potential buttressing effect (i.e. mechanical stress) that ice mélange would exert on the ice stream. Some other studies suggest that the presence of sea ice is only a reflection of oceanic conditions that are actually the dominating control on calving. While it seems clear that ice mélange may prevent calving, it is still not clear whether ice mélange has any (direct) effect in terms of buttressing.

We agree that the effect of ice mélange on calving remains an open question in glaciology, though a growing body of both observational and model evidence suggests that the effect is significant. We

have modified the introduction to explicitly state that this effect is yet to be fully established (P2L10).

Another important process is how crevasses propagate. The Nye approximation remains a simple approximation and has not (to my knowledge) been fully validated. The presence of basal crevasses upstream of grounding line is not seen in most of the radar echograms that I have seen. Basal crevasses start to form at the grounding line where tidal bending occurs, but (to my knowledge) are not visible under grounded ice. The authors present the calving law used here as "the truth", but it remains one possible (promising) representation of calving. I think the text should be less definitive in some places.

This statement seems to imply that our model predicts basal crevasses in grounded ice, but this is not the case. In fact, basal crevassing begins at the grounding line (Fig. 6b, white line) due to bending, just as described by the reviewer. To clarify this, we have modified the text (P11L18) to state that the southern side of the terminus is floating. We have also added a statement to justify our use of Nye as opposed to LEFM (P4L35). Finally, in the conclusions, we have modified the statement on the importance of basal crevasses (P12L16) to indicate that this is only what the model predicts, rather than a definite fact.

I obviously also have to say a few words about the comparison with Morlighem et al. 2016. The study presented here is undeniably more sophisticated but I still think that the authors have not rigorously demonstrated that a depth integrated model would respond differently. It is very difficult to compare the model from Morlighem et al. 2016 and the one presented here: they have a different initial state (this one is relaxed, so it may start from a different surface height), we use different meshes, possibly a different bed, different boundary conditions (I did not model lateral friction), etc. For example, I did not account for the presence of mélange: the inversion of basal drag is therefore expected to yield a slightly higher stress compared to the one of this model since the ice front includes here a stronger back stress from ice mélange. This will have an impact on the model sensitivity: my model having a stronger basal stress will be more stable. While both studies agree on many points (e.g., overall stability of Store, strong control of the bed geometry, etc), it is very difficult to disentangle why the models require different melt rates to be dislodged from their current position. We cannot claim that the model presented here is "better" simply because it is based on Full-Stokes. It is not what is shown. The only way to show that would be to collapse the model and run it with an SSA approximation (Elmer has this capability): that would make it possible to compare apples with apples. It would be great if this could be tested, but if this is not possible, the text needs to be less definitive in places (e.g., use conditional instead of present tense). ISSM also has a full-Stokes solver and from my experience, the results have almost always been very similar to results obtained with SSA. That being said, this was for vertical calving fronts.

We agree that a rigorous 'Calving MIP' would be a useful and illuminating exercise. Given the effort that would be required to reformulate our calving model to operate with SSA, we believe this is beyond the scope of the present study. We also agree, therefore, that we have not *rigorously* shown that full-Stokes is required to properly model calving. However, in the present study and in Todd et al. (2018), we have demonstrated the importance of several factors which cannot be properly represented in SSA: the grounding line transition, vertical bending due to buoyant forces, ice cliff force imbalance, and undercutting by submarine melting (i.e. non-vertical fronts). Conversely, we argue that similar behaviour in ISSM does not undermine the importance of full-Stokes unless the calving law, frontal boundary conditions, grounding line dynamics and submarine melting are adapted to take advantage of the full-Stokes 3D solution.

We have tempered our conclusions with regards to SSA by changing 'is likely' to 'may be' on [P8L11], and removing the final sentence of this paragraph: "However, the inability of the ISSM

model to reproduce the present-day terminus position highlights the need for fully 3D calving models equipped with physically-based calving laws." We have also modified P11L26 'it would not' to 'it may not', and P12L27 'will fail to capture' to 'may fail to capture'. Finally, we have added a couple of sentences to the end of the discusion [P11L36] to indicate that we expect long-term predictions of calving glaciers and ice sheets will be derived from models implementing simpler physics (e.g. SSA, higher-order approximations), and that we hope that insights from 3D full-Stokes simulations can feed into those models.

Finally, I found a bit unfortunate that the model breaks as soon as something interesting happens. I am surprised that reducing the time step does not fix the problem. Is there any possible way to improve the stability of the numerical implementation? At the end of the day, we want to be able to model ice front retreat, not just demonstrate its stability… Again, I think this is a great piece of work, I just think that a few things need to be put in perspective and some statements need to be nuanced.

Yes, it is annoying that the model breaks following significant retreat. This is a remeshing issue – we are planning to completely overhaul the remeshing algorithm, which will make the model significantly more robust. In response to this point, and a comment from the other reviewer, we have expanded the explanation of this problem in Section 2.2 [P4L24].

**2 Specific comments**

• title: the title is too generic. This paper is about one specific glacier, Store, I am not sure why the title says "Sensitivity of calving glaciers", it should be "Sensitivity of Store Glacier to ..."

We have changed the title of the article to "Sensitivity of a calving glacier to ice-ocean interactions..."

• p1 l38: Figure 2 is referenced after figure 1 (only mentioned page 2 line 26), maybe the order of the figures should be changed

Good point – order changed.

• p3 l24: 162 m (space between number of m, also l37 and in other places)

Fixed, thanks.

• p7 l20: this statement is misleading, ISSM is not "depth-averaged". We implemented several ice flow models in ISSM, including SSA and full-Stokes. In Morlighem et al. 2016, the depth-averaged SSA model was using, but ISSM itself is not depth-averaged.

Apologies – we were not aware of this. We have updated the text accordingly (P8L6), to state instead that "In previous work, the Ice Sheet System Model (ISSM) has been used with a vertically-integrated SSA solver…".

• p9 l9: -400 m a.s.l. sounds a bit awkward, maybe replace by 400 below sea level?

Good idea – done.

• p9 eq3: the hydrostatic imbalance quantity is difficult to appreciate. I think using the height above floatation (multiplying this quantity by H $\rho$i) would make it easier to evaluate how much the ice has to thin to be floating. The figure should also be adjusted accordingly.

We are purposefully trying to distance this analysis from the concept of height above flotation, because we know from the model (and observations) that the hydrostatic assumption is incorrect over short horizontal scales. Much of the ice in the southern side is significantly *below* flotation (hence the upward bending force), and the grounding line does not coincide with the HI=0 contour. In other words, over the spatial scales relevant to greenlandic outlet glaciers (few kms), height above buoyancy does *not* accurately predict the location of the grounding line. We have added a statement [P10L36] explaining the relationship between hydrostatic imbalance & height above buoyancy, and noting that this does not accurately predict grounding/flotation.

Looking into this further, we realise we have the wrong equation for what is shown in Figure 6c. It should read: $1 + (z_b/H)(\rho_{sw}/\rho_i)$. We have fixed this.

• p19 figure 3: all subplots have -2.132e6 at the top of the y axis, it is probably the offset for y, but it is not clear... It is a bit confusing, I am not sure it is necessary.

Yes, this is the coordinates in rather unwieldy NSIDC Sea Ice Polar Stereographic North. We have reworked this figure based on comments by the anonymous reviewer, and have replaced the x/y coordinate scale with a scale bar instead.